# Evaluating Effectiveness of YouTube Videos for Teaching Medical Students CPR: Solution to Optimizing Clinician Educator Workload during the COVID-19 Pandemic

**DOI:** 10.3390/ijerph18137113

**Published:** 2021-07-02

**Authors:** Osamu Nomura, Jin Irie, Yoonsoo Park, Hiroshi Nonogi, Hiroyuki Hanada

**Affiliations:** 1Department of Emergency and Disaster Medicine, Hirosaki University, Hirosaki 036-8562, Japan; jin-irie@hirosaki-u.ac.jp (J.I.); hanada68@hirosaki-u.ac.jp (H.H.); 2Harvard Medical School, Harvard University, Boston, MA 02114, USA; YSPARK@mgh.harvard.edu; 3Japan Resuscitation Council, Tokyo 151-0053, Japan; nonogijpulse@gmail.com

**Keywords:** YouTube, CPR, medical students, education, COVID-19 pandemic

## Abstract

(1) Background: This study aimed to evaluate the effectiveness of using a pre-existing video on CPR to support preclinical resuscitation education for medical students; (2) Methods: In total, 129 students selected to learn CPR using a pre-existing YouTube video or the conventional screencast video by their university faculties. All students responded to the pre- and post-training multiple-choice questionnaire on the basic knowledge of CPR, and, based on their responses, an analysis of covariance (ANCOVA) was conducted to assess the comparability of effectiveness across learning modalities. (3) Results: Among the students, 49 (38.0%) students selected the YouTube video to learn about CPR and were treated as the intervention group. The mean pre-test scores and post-test scores of the YouTube and the instructor’s video groups were 6.43 and 6.64, and 9.06 and 9.09, respectively. After controlling for the pre-test score effects, the results of ANCOVA did not show statistically significant differences between groups (*p* = 0.927), indicating comparable performance between groups that used YouTube and the instructor’s videos. (4) Conclusion: Utilizing YouTube videos is a useful teaching strategy for teaching CPR knowledge, which would reduce the burden on faculty of creating screencast lecture videos for online learning on resuscitation.

## 1. Introduction

Primary care and emergency physicians in academic institutions are reportedly facing a dilemma between clinical practice and teaching during the COVID-19 pandemic [1]. As social distancing is strongly recommended, in-person educational activities for medical students have been suspended, and the modality of preclinical learning for medical students has changed to an online modality [2]. These circumstances have prompted clinical educators to prepare and develop new educational content for online teaching about emergency medicine and to record lecture videos [3]. However, due to the shortage of required medical personnel during the pandemic, primary care and emergency physicians have had to increase their clinical work hours and have little time to prepare for preclinical education of medical students [4]. Therefore, the educational burden on academic physicians, in addition to their clinical workload, might compromise their wellbeing during the COVID-19 pandemic [5,6].

One potential solution to this problem is to use existing Free Open Access Meducation (FOAMed) resources, such as online videos, to supplement emergency medicine course content for medical students [7,8]. The field of resuscitation education has rich experience with the use of instructional video resources [9,10]. Videos explaining and demonstrating standardized resuscitation protocols have long been used in cardiopulmonary resuscitation (CPR) courses. While previous studies reported that the quality of some of the FOAMed resources is inadequate [11,12], new criteria ensuring the educational quality of such materials have been established and used to optimize the educational opportunities for trainees even during the COVID-19 pandemic [13,14,15]. Therefore, the application of quality-assured FOAMed CPR videos for preclinical resuscitation training of medical students has the potential to be an alternative instructional methodology during the COVID-19 pandemic.

This study aimed to evaluate the effectiveness of using YouTube videos on CPR to complement preclinical resuscitation education of medical students. We hypothesized that FOAMed videos available on YouTube might be as effective as newly created screencast videos by medical school faculties of basic CPR knowledge to medical students. The rationale of this research is to indicate the practical evidence of using YouTube videos can reduce the academic burden on primary care and emergency physicians with academic responsibilities of teaching medical students during the COVID-19 pandemic.

## 2. Materials and Methods

### 2.1. Study Design and Participants

This was an experimental educational study including 4th-year medical students enrolled in a 6-year-program at Hirosaki University, Japan, that evaluated the emergency medicine course on resuscitation for 4th-year medical students. The participants had previously studied basic and clinical medicine in a didactic lecture-based medical education program and were preparing for a clinical clerkship the following year. Among the 134 eligible students, 5 were excluded due to missing data, and the remaining 129 students were included in the analysis (Figure 1).

### 2.2. Study Setting

The resuscitation module, which included the contents of basic life support (BLS) and advanced life support (ALS) [9], was taught online in response to the COVID-19 pandemic, and students learned the necessary information by watching the 70-min screencast lecture videos of the university instructors. For this program, we prepared the BLS teaching content for a layperson, including one of two methods of delivery: an institutional screencast lecture video created by the course instructor (ON) or a pre-existing YouTube video (15 min each), both of which cover the same teaching contents of CPR.

### 2.3. Study Protocol

The students selected the delivery method (i.e., institutional screencast or YouTube) based on their learning preference. No students were allowed to watch both videos or switch from one video to the other. All 129 students first watched the introductory part (15 min) of the module after responding to the pre-training quiz (10 min), and then learned about BLS for laypersons by selecting their preferred delivery method (institutional screencast or YouTube video: 15 min). They finally completed the post-lecture quiz (10 min) after watching the other contents (i.e., BLS for healthcare providers and ALS: 20 min). As the study outcome, we assessed students’ performance using the pre-and post-lecture quizzes, including the same 10 multiple-choice questions evaluating the knowledge of BLS, and its maximum possible score was 10 (Appendix A). No clinical CPR skill assessments were conducted due to the COVID-19 pandemic situation.

### 2.4. Intervention (YouTube)

The videos watched by Students who preferred to watch the YouTube videos included two videos that were initially created in English by Zoll Singapore to teach CPR to laypersons, and were then translated into Japanese by Asahi Kasei ZOLL Medical Corporation [16,17]. The Japan Resuscitation Council (JRC) subsequently evaluated the Japanese videos in concordance with the 2015 JRC resuscitation guideline and recommended their use for learning CPR by uploading them on YouTube and on the Council’s FOAMed Webpage for CPR advocacy [18].

These YouTube videos were selected by the course instructor for this study, as they covered the same content as the screencast video by the instructor and the contents included the key aspects of CPR education that are reported in the literature, namely: (1) assessing scene safety, (2) checking victim responsiveness, (3) initiating contact with emergency medical services, (4) proper hand positioning for CPR, (5) accurate compression rate (100–120 per minute), and (6) appropriate chest compression depth (2–2.5 inches) [19]. The YouTube videos were not modified for the purpose of the course.

### 2.5. Data Analysis

An analysis of covariance (ANCOVA) was conducted following the standard protocol, using post-test scores as the dependent measure and the pre-test score as covariates. The group of students who chose the YouTube video was treated as the intervention group. Statistical analyses were conducted using SPSS version 23.0 (IBM Corporation, 2018, Armonk, NY, USA).

### 2.6. Ethical Considerations

This study was exempted from review by the Ethics Committee of Hirosaki University, Graduate School of Medicine. Consent from the students for study participation was obtained on an opt-out basis.

## 3. Results

Of the 129 students, 49 (38.0%) students selected the YouTube video to learn about BLS and were treated as the intervention group (Figure 1), and 80 (62.0%) students selected the institutional screencast by the instructor. The baseline mean scores of the YouTube and instructor’s video groups were 6.43 (standard deviation [SD] = 1.54) and 6.64 (SD = 0.92), respectively. Post-test scores were 9.06 (SD = 0.92) and 9.09 (SD = 1.37), respectively (Table 1). Assumptions for parallel slopes were met, taking into consideration the fact that the interaction between the group and pre-test score was not significant (*p* = 0.984).

Controlling for the pre-scores’ effect, ANCOVA results did not show statistically significant differences (*p* = 0.927), indicating that the effectiveness of the YouTube video and the instructor’s video was similar, thereby suggesting comparability of performance between the two learning modalities (Figure 2).

## 4. Discussion

This study suggests that the educational impact of using YouTube videos for teaching CPR knowledge is comparable to that of using newly created screencast videos by medical school faculties, while simultaneously reducing the burden on faculty in terms of the need to create screencast lecture videos on resuscitation for online learning. This finding indicates that medical school faculties can improve their wellbeing using FOAMed resources effectively to optimize the educational workload.

FOAMed, such as YouTube videos, have been reported to be useful resources for teaching BLS and CPR, which is compatible with contemporary guidelines [11,20]. The benefits of FOAMed resources are their content variety, flexibility, and accessibility [8]. For example, YouTube videos can expand the students’ learning opportunities even outside the campus. Besides, the visual and auditory effects of video resources in FOAMed facilitate learners’ acquisition and retention of knowledge [7]. Additionally, YouTube is reported to be the most commonly used resource for medical trainees to learn basic procedural skills and satisfies the learning preference of millennials [21,22]. Although it is still arguable whether the quality, accuracy, and updated content of the FOAMed resources are appropriately monitored [11,12], measurement tools evaluating the quality of FOAMed resources have been developed and have the potential to address their rigor, reliability, and efficacy [13,14,15]. However, it has still not been clarified whether using existing YouTube videos on CPR can serve as an alternative to creating lecture videos for preclinical medical students. In this study, we found that a BLS YouTube video, which was carefully reviewed using the established methodology, might be as effective as an instructor-created screencast video for teaching basic CPR knowledge to medical students. This could be a significant solution for reducing the workload of emergency medicine instructors and improving their emotional wellness during the current pandemic [23,24].

The direct impact of our intervention may be modest, as the academic setting of the study was to provide preclinical medical students with the basic knowledge of CPR for laypersons; however, we started this study with minimum intervention in view of the possible ethical problems that a more thorough on-going intervention might have created.

Several limitations of this study should be considered. First, the participants were not strictly randomized into two groups as they were grouped based on their learning preference, resulting in the risk of selection bias and inequality in the number of participants in the groups. Second, we assessed the students’ performance by focusing on their knowledge acquisition on CPR; thus, the educational impact of the examined resources on practical skill acquisition is still uncertain. Third, we did not assess the long-term outcomes of our intervention. However, a long-term study has the risk of including many confounding factors that might influence the student’s long-term outcomes, which makes it difficult to assess the educational impact of our course on the students’ performance. Therefore, we designed this study with a short-term focus. Fourth, this study examined the medical students’ knowledge on “CPR for laypersons”, not “for healthcare providers”. This might limit the generalizability of the findings; however, essential aspects of the CPR procedure are the same regardless of whether the content is created for laypersons or healthcare providers, and we thus believe that “CPR for laypersons” is still an important topic for preclinical medical students. Finally, we did not assess the students’ actual behaviors while watching the videos; thus, it was possible for them to skip certain parts of the videos or watch the videos at a faster speed (e.g., 1.25 speed). This limitation can be addressed in future research using the predefined protocol that carefully monitors students’ behavior on watching YouTube.

## 5. Conclusions

Existing YouTube videos can be as effective as newly created screencast videos by medical school faculties in providing the necessary knowledge about CPR to preclinical medical students. Utilization of YouTube CPR videos can be an alternative instructional methodology during the COVID-19 pandemic.

## Figures and Tables

**Figure 1 ijerph-18-07113-f001:**
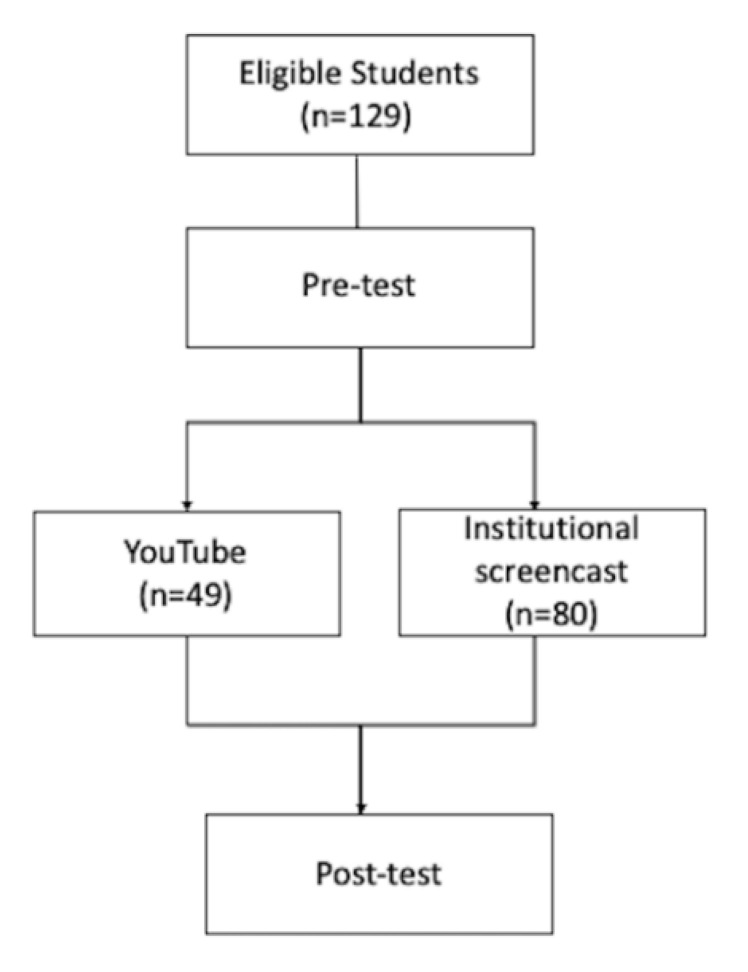
Study flow chart.

**Figure 2 ijerph-18-07113-f002:**
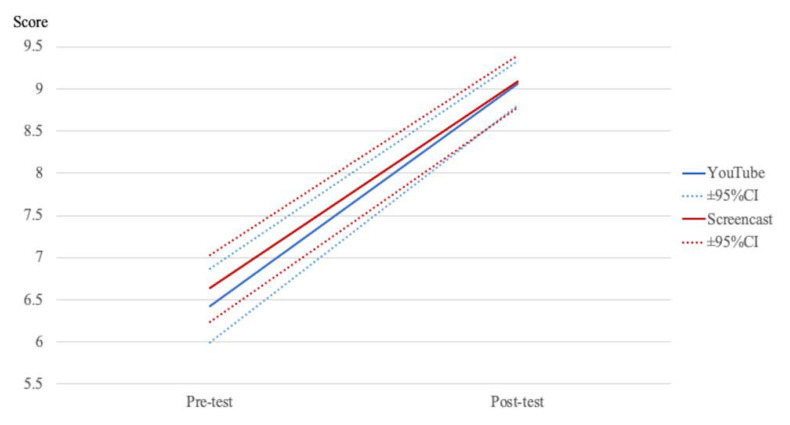
Results of the analysis of covariance.

**Table 1 ijerph-18-07113-t001:** Descriptive statistics of the pre and post-tests in both groups.

Type	Test	*n*	Mean	SD
YouTube	Pre-Test	49	6.43	1.54
	Post-Test	49	9.06	0.92
Screencast by Instructor	Pre-Test	80	6.64	1.77
	Post-Test	80	9.09	1.37
Overall	Pre-Test	129	6.56	1.69
	Post-Test	129	9.08	1.22

## Data Availability

The data that support the findings of this study are available from the corresponding author upon reasonable request.

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
