# Peer review of "Evaluating Effectiveness of YouTube Videos for Teaching Medical Students CPR: Solution to Optimizing Clinician Educator Workload during the COVID-19 Pandemic"

_ijerph, 2021, doi:10.3390/ijerph18137113_

Round 1

Reviewer 1 Report

The authors have addressed all of my comments from the previous round of reviews. I have no further comments.

In my opinion, a comment on behaviours which were unsupervised in the YouTube group (skipping sections, watching at 1.25X speed) are warranted. 

Author Response

Comment

"In my opinion, a comment on behaviours which were unsupervised in the YouTube group (skipping sections, watching at 1.25X speed) are warranted. "

Response

Thank you for your feedback. In response to your comment, we added a description regarding this limitation. "This limitation can be addressed in future research using the predefined protocol that carefully monitors students' behavior on watching YouTube." (Line172-174)

Reviewer 2 Report

The manuscript is well written and conforms to the requirements for this type of paper. However, the rationale for the study is not justified as it does not add value to the knowledge already available. There is no reason to think that two videos with equivalent content and duration, and supervised quality would have different effects on learning simply because they come from different sources (institutional screencast video by the instructor vs. YouTube). Likewise, the YouTube videos use to reduce instructor overload is not a hypothesis to be tested, but a reality that does not require demonstration. The authors themselves cite in their paper other studies in which the usefulness of YouTube videos for teaching CPR has already been demonstrated.

Author Response

Thank you very much for your time in reviewing our paper.

 We do appreciate your comment that our manuscript meets the requirements for the type of paper. While we understand your comments that this article may not add the new findings to the body of knowledge in the literature, the other two reviewers provided very positive comments for the publication. Thus, we believe that our study's finding might be valuable for some readers, such as clinician-educators who struggle to optimize the balance between the clinical and educational workload during the COVID-19 pandemic.
 We appreciate your time and effort in reviewing our paper, and your comments enabled us to improve our paper. 

Reviewer 3 Report

It is a properly performed study, comparing CPR Youtube Videos to videos produced by medical faculties, demonstrating that some Youtube videos did not result in inferior knowledge. 

the videos aim at the evaluation of the theoretical knowledge of the students, and I think that it should be made very clear, that this is only an adjunct tool to practical training using a resuscitation doll, as finding the right pressure points and depth of sternal impressions may be more tricky in practice than in theory.

I would also mention the usually better availability of YouTube videos for training, even outside a campus. 

After all it is a valuable part of the toolbox in resuscitation training. 

Author Response

Comment1

"videos aim at the evaluation of the theoretical knowledge of the students, and I think that it should be made very clear, that this is only an adjunct tool to practical training using a resuscitation doll, as finding the right pressure points and depth of sternal impressions may be more tricky in practice than in theory."

Response

Thank you for your comment. We added the descriptions that this study outcome focused on acquiring CPR knowledge in the method and limitation sections.

"As the study outcome, we assessed students' performance using the pre-and post-lecture quizzes, including the same ten multiple-choice questions evaluating the knowledge of BLS, and its maximum possible score was 10. No clinical CPR skill assessments were conducted due to the COVID-19 pandemic situation."( Line83-86)

Second, we assessed the students' performance focusing on their knowledge acquisition on CPR; thus, the educational impact of the examined resources on practical skill acquisition is still uncertain." (Line 159-161)

We also changed other confusing descriptions using "CPR skills" to "CPR knowledge" These parts were marked with YELLOW.

Comment 2

"I would also mention the usually better availability of YouTube videos for training, even outside a campus. "

Response

Thank you for your supportive comment. I agree with you and added the description of better availability of the YouTube videos in the discussion.

"For example, YouTube videos can expand the students’ learning opportunities even outside the campus." (Line 137-138).

Round 2

Reviewer 2 Report

I have no additional comments for the authors.

Author Response

This manuscript is a resubmission of an earlier submission. The following is a list of the peer review reports and author responses from that submission.

Round 1

Reviewer 1 Report

The utilization of YouTube contents as means of online education in this Pandemic era to supplement face-to-face classes for med students is an appropriate direction to take. The aim of the current study was to investigate whether such YouTube education is effective. However, this study has following problems that need to be addressed: 

  1. Video-made education and YouTube education are only different in terms of whether they are made by the researchers or the school, and are not different in their contents or methods. In other words, apart from the cast of the videos, there is no evidence to think that there will be any differences in their effectiveness.

  1. The limitation of this study is that, even if YouTube education is more effective than video-based education, it is unknown where the difference comes from.

  1. The YouTube video contents are already verified in JCR, and the quality is acknowledged.

  1. Except that there is no significant difference, any new information that can be obtained from reading this paper is that YouTube can be used for education. There is no new information on why, or which conditions the YouTube contents need to fullfil for its effectiveness, etc.

Author Response

Comments and Suggestions for Authors

  1. "Video-made education and YouTube education are only different in terms of whether they are made by the researchers or the school, and are not different in their contents or methods. In other words, apart from the cast of the videos, there is no evidence to think that there will be any differences in their effectiveness."
  1. "The limitation of this study is that, even if YouTube education is more effective than video-based education, it is unknown where the difference comes from."

Response:

Thank you for your comments. Our unclear descriptions might have confused you.

The screencast video was made by the researcher (i.e., instructor), and the YouTube video was created by a pharmaceutical company (ZOLL Medical Corporation) not by the school.

We do not mean that YouTube education is more effective than video-based education, but that the educational effectiveness of both methods is comparable. However, the course instructor's burden can be reduced using the YouTube strategy, because it is unnecessary for the instructor to create a new screencast video for preparing the course materials, which is the benefit of using YouTube education. In summary, we suggest that a YouTube video is useful because it can provide the requisite knowledge with comparable educational benefits for students, while reducing the educational burden on the instructor to create course materials.

To clarify these points, we added relevant descriptions related to our hypothesis (in the introduction: Lines 59-64) and interpretation of the results (in the discussion: Lines 143-146). We also reorganized the structure of the methods by adding the headings of “2.2. Study setting (Line 74)”, “2.3. Study protocol (Line 83)”, “2.4. Intervention (YouTube: Line 94)”, and “2.6 Ethical considerations (Line 115)”. Finally, we avoided use of the words "effectiveness" or "effective" as much as possible in the manuscript.

  1. "The YouTube video contents are already verified in JCR, and the quality is acknowledged."
  1. "Except that there is no significant difference, any new information that can be obtained from reading this paper is that YouTube can be used for education. There is no new information on why, or which conditions the YouTube contents need to fulfill for its effectiveness, etc."

Response:

Thank you for your comments. I have added descriptions on the educational benefits of using YouTube videos (Lines 151-153).

Reviewer 2 Report

The presented study evaluates the effectiveness of YouTube videos to train students virtually, a topic which has become of dire importance during the pandemic.

The authors conclude that YouTube videos (a) are an effective teaching method (b) can reduce burden on faculty. They used FOAMed (Free Open Access Meducation) resources….It’s not entirely clear if these FOAMed resources are the content that was uploaded to YouTube or how it was modified prior to being uploaded to YouTube.

Sample size of 129 was sufficient for the conclusions mad. Group 1 followed the content created by the course instructor. Group 2 watched a 15 minute YouTube video (lines 69-72).

Allowing students to choose between teaching methods was an interesting choice, and potentially limits conclusions which the authors discuss. I would have been interested in a deeper comparison between the content in both interventions. What knowledge was included in the instructor content which was left out in the YouTube intervention (and vice-versa). 

Would a student who is well versed in CPR be more likely to choose YouTube because they can skip over content? Or would they consider listening to the content at 1.25 speed (these are available by looking at YouTube metrics). 

More information is required on the content of pre/post test scores. Possible consider including in an appendix. 

I felt the paper was great, but clarifications as noted above would be appreciated.

Author Response

Comment 1

"The authors conclude that YouTube videos (a) are an effective teaching method (b) can reduce burden on faculty. They used FOAMed (Free Open Access Meducation) resources….It’s not entirely clear if these FOAMed resources are the content that was uploaded to YouTube or how it was modified prior to being uploaded to YouTube".

Response:

Thank you for your comments. For training on BLS, students who chose to watch the YouTube videos were instructed to access the JRC FOAMed Web page in which YouTube videos were embedded. Thus, we did not modify the videos. We clarified this in the Materials and Methods section (Lines 95-108 and Line 108).

Comment 2

"Sample size of 129 was sufficient for the conclusions made. Group 1 followed the content created by the course instructor. Group 2 watched a 15 minute YouTube video (lines 69-72). Allowing students to choose between teaching methods was an interesting choice, and potentially limits conclusions which the authors discuss."

Response:

In response to your comments, we added arguments on selection bias in the limitations section (Lines 168-171). Since we understand that this is a significant limitation, we have downplayed the conclusion, stating that “YouTube might be as effective as” the screencast videos (Lines 187-190).

Comment 3

"I would have been interested in a deeper comparison between the content in both interventions. What knowledge was included in the instructor content which was left out in the YouTube intervention (and vice-versa). "

Response:

The instruction contents in both interventions were the same (basic CPR). We have clarified descriptions regarding this point in the Materials and Methods section (Line 81-82 and Lines 102-108).

Comment 4

"Would a student who is well versed in CPR be more likely to choose YouTube because they can skip over content? Or would they consider listening to the content at 1.25 speed (these are available by looking at YouTube metrics).

Response:

We did not assess if students who watched the YouTube videos behaved as you indicated. The YouTube videos are accessible to a wider audience in Japan, and the metrics might not be accurate enough to evaluate our students’ behavior. However, we consider that students were less likely to skip or watch the YouTube videos at 1.25 speed as they were informed that they were required to take the post-test at the end of the online class. We have added this as a study limitation (Lines 182-185).

Comment 5

"More information is required on the content of pre/post test scores. Possible consider including in an appendix. "

Response:

Thank you for your comment. We included the pre/post-test scores in the appendix.

Round 2

Reviewer 1 Report

This study examined the utilization of YouTube contents as means of online education in this Pandemic era. However, this study has following problems: 

  1. Video-made education and YouTube education are only different in terms of whether they are made by the researchers or the school, and are not different in their contents or methods. 
  1. The YouTube video contents are already verified in JCR, and the quality is acknowledged.
  2. The meaning of no difference between effect of YouTube and effect of Video is that the research does not show difference between the effect of two methods. There is no new information on which method are superior than other method.